# Anti-Inflammatory and Gut Microbiota Modulation Potentials of Flavonoids Extracted from *Passiflora foetida* Fruits

**DOI:** 10.3390/foods12152889

**Published:** 2023-07-29

**Authors:** Xiangpeng Han, Ya Song, Riming Huang, Minqian Zhu, Meiying Li, Teresa Requena, Hong Wang

**Affiliations:** 1Guangdong Provincial Key Laboratory of Food Quality and Safety, College of Food Science, South China Agricultural University, Guangzhou 510642, China; hanxp1996@163.com (X.H.); songya_1990@163.com (Y.S.); huangriming@scau.edu.cn (R.H.); zhuminqian@stu.scau.edu.cn (M.Z.); lmy1982@scau.edu.cn (M.L.); 2Guangdong Laboratory for Lingnan Mordern Agriculture, Guangzhou 510642, China; 3Instituto de Investigación en Ciencias de la Alimentación CIAL (CSIC), Campus UAM Cantoblanco, 28049 Madrid, Spain

**Keywords:** *Passiflora foetida* fruits, flavonoid-rich fraction, anti-inflammatory activity, gut microbiota, short-chain fatty acids

## Abstract

This study aimed to explore the anti-inflammatory and gut microbiota modulation potentials of flavonoid-rich fraction (PFF) extracted from *Passiflora foetida* fruits. The results showed that PFF markedly reduced the production of nitric oxide (NO), tumor necrosis factor α (TNF-α), and interleukin 6 (IL-6) in LPS-stimulated RAW 264.7 cells. Meanwhile, PFF treatment also effectively decreased the phosphorylation levels of MAPK, PI3K/Akt, and NF-κB signaling-pathway-related proteins (ERK, JNK, p38, Akt, and p65). Moreover, PFF had an impact on microbial composition and metabolites in a four-stage dynamic simulator of human gut microbiota (BFBL gut model). Specifically, PFF exhibited the growth-promoting ability of several beneficial bacteria, including *Bifidobacterium*, *Enterococcus*, *Lactobacillus*, and *Roseburia*, and short-chain fatty acid (SCFA) generation ability in gut microbiota. In addition, spectroscopic data revealed that PFF mainly contained five flavonoid compounds, which may be bioactive compounds with anti-inflammatory and gut microbiota modulation potentials. Therefore, PFF could be utilized as a natural anti-inflammatory agent or supplement to health products.

## 1. Introduction

Inflammation is a pivotal defensive response of the host immune system against injurious stimuli, such as toxic compounds, injured cells, pathogenic organisms, and irradiation [1,2]. It can prevent infection caused by early cell damage, eliminate damaged cells or tissue, and promote tissue healing in the human body [2]. However, excessive or uncontrolled inflammation may result in multiple health diseases, including inflammatory bowel disease (IBD), allergies, metabolic syndrome, diabetes, autoimmune diseases, cardiovascular diseases, and cancer [3]. Currently, most epidemiological studies have demonstrated that inflammation-related diseases are closely interrelated with changes in gut microbiota [4,5,6]. For instance, the gut microbiota composition of IBD patients exhibits specific characteristics in comparison with that of non-IBD patients [7]. Of note, certain microorganisms play vital roles in host homeostasis, such as defending against viral and pathogenic invasion, absorbing nutrients and energy, boosting the immune system, and improving metabolism [8,9]. To date, various purified or synthetic drugs have been explored to regulate and alleviate inflammation-related diseases, including immunomodulators, steroids, and non-steroidal anti-inflammatory drugs [2,10], whereas the above intervention drugs generally provide limited clinical effectiveness and have a series of adverse effects [5]. Therefore, the current research focuses on exploring and developing natural anti-inflammatory agents derived from medicine food homology plants or herbal medicines to enhance their efficacy and safety.

In recent years, evidence has been mounting that natural flavonoid-enriched foods, such as fresh fruits, vegetables, and tea, are helpful in decreasing the risk of inflammation-related diseases [11]. Generally, flavonoids possess excellent anti-inflammatory properties to effectively inhibit the inflammatory processes by disrupting inflammatory cytokine release and regulating inflammation-associated signaling pathways without harmful side effects [5]. Moreover, many studies have revealed that dietary intake of flavonoids is intimately relevant to the variety and composition of gut microbiota [12]. Significantly, improving the dysregulated gut microbiota is a momentous target of new therapies for intestinal inflammations [13]. For instance, flavonoids can attenuate the host inflammatory response and reduce the incidence or severity of IBD through interaction with the specific microbiota and their metabolites [14]. Moreover, flavonoids can also modulate gut microbiota by reducing pathogens (such as *Staphylococcus*, *Pseudomonas*, *Proteus*, etc.) and increasing probiotics (such as *Lactobacillus* and *Bifidobacteria*, etc.) [15], which can alleviate the inflammatory responses by reducing endotoxin production, thereby promoting gut health [16]. Hence, growing attention to dietary nutrition and illness prevention has increased the daily demands for flavonoid-enriched foods.

*Passiflora foetida* fruit (*Passifloraceae*) is an edible berry commonly consumed in tropical areas, such as Brazil and South China, for treating acute edema, traumatic cornea, conjunctivitis, and relieving asthma [17]. To date, the biological functions of *P. foetida* fruits, including antibacterial, anti-osteoporotic, anti-inflammatory, and antioxidant activities, have been reported [11]. Our previous studies showed that *P. foetida* fruits are mainly rich in amino acids (1097 mg/100 g), minerals (595.75 mg/100 g), sugars (3.34 g/100 g), organic acids (0.24 mg/100 g), and polyphenols (284.4 mg/100 g), especially flavonoids (239 mg/100 g), which are the major component of polyphenols [18]. Moreover, the flavonoids were another main component besides the polysaccharides in the aqueous extract of *P. foetida* fruits, and the polysaccharides have been characterized and demonstrated to possess excellent immune-enhancing activity [18,19,20]. However, the compositions and bioactive benefits of flavonoids from *P. foetida* fruits still need to be discovered. Therefore, this proposed study aims to explore the in vitro anti-inflammatory activity and gut microbiota modulation potential of PFF isolated from *P. foetida* fruits. This work will provide scientific evidence for new uses of *P. foetida* fruits and promote its development and application as a functional food for human health.

## 2. Materials and Methods

### 2.1. Materials and Reagents

The mature fruits of *P. foetida* were harvested from Dongfang city, Hainan province, China, in May 2021. RAW264.7 cells were purchased from American Type Culture Collection (ATCC, Manassas, VA, USA). Dulbecco’s Modified Eagle’s medium (DMEM) was obtained from Gibco BRL (Gaithersburg, MD, USA). Penicillin, streptomycin, and fetal bovine serum (FBS) were products of HyClone (Logan, UT, USA). Lipopolysaccharide (LPS), 3-(4,5-dimethylthiazol-2-yl)-2,5-diphenyltetrazolium bromide (MTT), sulfuric acid, trifluoroacetic acid (TFA), enhanced chemiluminescence (ECL) kit, AB-8 macroporous resin, and polyamide filler were produced by Yuanye Biotechnology Co., Ltd. (Shanghai, China). Bicinchoninic acid (BCA) kit, nitric oxide (NO) detection kit, ELISA kits (TNF-α and IL-6), and 4′,6-diamidino-2-phenylindole (DAPI) were purchased from Beyotime Biotechnology Co., Ltd. (Shanghai, China). Sephadex LH-20 dextran gel was purchased from General Electric Co. (GE, Fairfield, CT, USA). The methanol and acetonitrile were produced by Sigma Chemical Co. (St. Louis, MO, USA).

### 2.2. Preparation of PFF

*P. foetida* fruits were separated and washed manually with distilled water, then dried in an oven. The dried samples were ground and filtrated to prepare *P. foetida* powder, which was preserved in an oxygen barrier bag at −20 °C until further use. PFF extraction and isolation were performed according to a previous study [21]. Briefly, the crude aqueous extract of *P. foetida* (AEP) was obtained via infusion with water (65 °C, powder: solvent, 1:10, *w*/*v*) for 2 h. The concentrated AEP was precipitated by adding ethanol (96%) overnight at 4 °C, then loaded on AB-8 macroporous resin column, eluted sequentially through distilled water and extracted with ethanol (80%). Subsequently, the extract was further purified by a polyamide column to obtain PFF.

### 2.3. Cell Culture and Viability Assay

Cell viability was determined using the MTT method [2]. RAW264.7 cells (200 µL) were inoculated into a 96-well microplate (1 × 10^5^ cells/mL) and cultured in DMEM (containing 10% FBS and 1% penicillin-streptomycin) at 37 °C for 24 h. The medium was replaced with fresh DMEM containing various concentrations of PFF (0, 6.25, 12.5, 25, 50, 100, 200, 400, and 600 μg/mL). After incubation, the medium was removed, and 200 µL of MTT solution (0.5 mg/mL) was added to a 96-well plate, which was cultured for 4 h at 37 °C. Subsequently, the plate was washed with sterile PBS, and 150 µL of DMSO was added to each well. After shaking in the dark for 15 min, the absorbance was recorded at 570 nm by a microplate analyzer (Molecular Device, San Jose, CA, USA).

### 2.4. Determination of NO and Cytokines

The production of NO, TNF-α, and IL-6 was determined based on previously reported methods [9]. In brief, RAW264.7 cells were inoculated into a 48-well plate (1 × 10^5^ cells/mL) containing LPS (1 µg/mL) and PFF (0, 50, 100, and 200 μg/mL). After 24 h of incubation, the supernatant was harvested to quantify NO, TNF-α, and IL-6 levels. NO production was determined using Griess reagent, and equal volumes of culture supernatants and Griess reagent were mixed and reacted for 15 min. Then, OD_540_ was determined by a microplate analyzer. Meanwhile, the production of IL-6 and TNF-α was determined using ELISA kits.

### 2.5. Western Blot Analysis

Western blot was performed based on a previously described method [22]. In short, RAW 264.7 cells were inoculated into a 6-well plate (1 × 10^5^ cells/mL) and treated with LPS (1 µg/mL) and PFF (0, 50, 100, and 200 μg/mL) for 24 h. After incubation, the cells were harvested and lysed in radioimmunoprecipitation assay (RIPA) buffer. After centrifugation (12,000× *g*, 4 °C, 20 min), the protein concentration of the supernatant was quantified by BCA Kit. Meanwhile, the samples and ×5 loading buffer were mixed and heated (100 °C, 10 min) to denature the protein completely. Then, 50 μg of total protein was separated by SDS-PAGE gel and transferred to a polyvinylidene difluoride membrane, which was blocked by 5% BSA and incubated with the primary antibodies. After that, membranes were further incubated with the secondary antibody. Finally, protein bands were reacted with ECL solutions and visualized using an imaging system (GE, Chicago, IL, USA).

### 2.6. Immunofluorescence Analysis

Confocal microscopy was performed as described by Duan et al. [23]. RAW264.7 cells (1 × 10^5^ cells/mL) were pretreated in a 6-well plate with LPS (1 μg/mL) and PFF (0, 100, and 200 μg/mL). After that, cells were collected and washed gently using sterile PBS, sequentially fixed with 4% formaldehyde for 30 min, and permeabilized with 1% Triton X-100 for 30 min. Afterward, fixed cells were blocked by 5% BSA, washed and incubated with p65 antibody overnight. Subsequently, the cells were incubated with FITC-labeled second antibody in darkness for 2 h, then stained with DAPI in darkness for 30 min. Images were observed using confocal laser scanning microscopy (Leica, Wetzlar, Germany).

### 2.7. Dynamic Simulator of the Gut Microbiome

The preparation of fecal samples was carried out according to a method reported by Tang et al. [24]. Specifically, fresh fecal samples were collected from five healthy human volunteers (BMI between 18.5 and 24.9 kg/m^2^) who did not have intestinal diseases or had not taken antibiotics for the previous three months and followed a flavonoid-free diet for three days before fecal collection. The fecal samples were immediately stored in an anaerobic tank and then diluted in sodium phosphate buffer (0.1 M, pH 7.0) with sodium thioglycolate and homogenized to obtain a 20% (*w*/*v*) feces slurry.

The BFBL gut model (Appendix A) is a four-stage reactor system that simulates the small intestine (SI), ascending colon (R_1_), transverse colon (R_2_), and descending colon (R_3_) of humans in vitro [25]. Briefly, three colon reactors (R_1_, R_2_, and R_3_) were first filled with nutritive medium (pH 2.0) in a volume of 250, 400, and 300 mL, respectively. Subsequently, each colon reactor was inoculated with 20 mL of feces slurry, then continuously stirred (150 rpm) with an Agimatic N magnetic stirrer with heating (Selecta, Spain), and the temperature was maintained at 37 °C. Meanwhile, the system was maintained as anaerobic by continuously flushing N_2_, and pH values of the colonic reactors were automatically adjusted to maintain 5.7 ± 0.2 in the R_1_, 6.3 ± 0.2 in the R_2_, and 6.8 ± 0.2 in the R_3_.

To maintain gut homeostasis, the small intestine (SI) was fed 75 mL of nutrient medium (pH 2.0) and 40 mL of artificial pancreatic juice mixture (6 g/L cholangium, 1.9 g/L porcine pancreatic trypsin, and 12.5 g/L NaHCO_3_) three times per day for two weeks. Small intestine digestion was conducted at 37 °C, and vessel contents were automatically transferred to the following colonic compartment, which activated the transport of colonic content between R_1_, R_2_, and R_3_ reactors regulated by level sensors to maintain a constant reactor volume. During the feeding process, all compartments were flushed with nitrogen gas to maintain an anaerobic state. After a two-week stabilization period (CK), the BFBL gut model was subjected to a one-week intake period by adding PFF (4 g/L) to the nutritive medium, followed by a one-week washout period (WO) with the addition of nutrient medium. Finally, samples were collected from three colon reactors daily throughout the study period and preserved at −20 °C until use.

### 2.8. Microbiota Composition Analysis

Genomic DNA of microbial samples from R_1_, R_2_, and R_3_ reactors was extracted as previously reported by Requena et al. [25]. The gene-specific primers, standards, amplicon size, and amplification conditions for the quantification of targeted microbial groups are listed in Appendix A. The SYBR Green-based RT-qPCR assay was conducted to quantify DNA samples and standards using a ViiA7 Real-Time PCR System (Thermo Fisher Scientific, Waltham, MA, USA).

### 2.9. SCFAs Analysis

The BFBL gut model samples (1 mL) were collected daily from each reactor (R_1_, R_2_, and R_3_) for SCFAs analysis during each period. SCFAs were quantified by a method reported by Barroso et al. [26]. In short, the filtered (0.22 μm) samples were analyzed by an HPLC system equipped with Rezex ROA-Organic Acids column (300 mm × 7.8 mm, Phenomenex, Macclesfield, UK) and UV-975 detector (Jasco, Tokyo, Japan). The mobile phase was 0.005 M sulfuric acid solution with a linear gradient of 0.6 mL/min, and elution curves were monitored at 210 nm. Calibration curves were established for SCFAs with concentrations ranging from 1 to 100 mM.

### 2.10. Isolation of the Main Flavonoids in PFF

The extraction and separation of the main flavonoids in PFF were conducted based on previous studies [27,28], with slight modifications. Firstly, PFF was purified by permeation on the Sephadex LH-20 column using methanol as the mobile phase (0.5 mL/min). Subsequently, the collected fraction was subjected to gradient preparative liquid chromatography, conducted using the Prep 150 LC set with a 2988 photodiode array detector, and connected to a SunFire C18 OBD Prep column (250 mm × 19 mm, 5 μm, Waters, Milford, MA, USA) using 80% methanol as an eluent (5.0 mL/min).

### 2.11. Initial Analysis of the Main Flavonoids in PFF

The initial analysis of the main flavonoids in PFF was conducted as previously described by Cho et al. [29] and Chao et al. [30]. Briefly, each purified sample was dissolved in methanol (1 mg/mL), then sample films were dotted on potassium bromide crystal tablets by the capillary. The compressed tablets were scanned by Fourier transform infrared spectrometry (Bruker, Rheinstetten, Germany) in a frequency range of 4000–400 cm^−1^, and the infrared absorption spectra of each purified sample were recorded. Simultaneously, they were also diluted and scanned from 200 to 600 nm using UV-Vis spectrophotometry (Thermo Fisher Scientific, Waltham, MA, USA). Furthermore, liquid chromatography–mass spectrometry (LC-MS) analysis was performed using LTQ Orbitrap mass spectrometry (Thermo Fisher Scientific, Waltham, MA, USA).

### 2.12. Statistical Analysis

One-way ANOVA was carried out to assess statistical significance using SPSS 20.0 (IBM, Armonk, NY, USA). All graphical evaluations were conducted by GraphPad Prism 8.0 (GraphPad, San Diego, CA, USA).

## 3. Results and Discussion

### 3.1. Effects of PFF on the Level of NO and Cytokines Produced in RAW264.7 Cells

Macrophages are the most important immune cells in various immune responses, especially in the immune response; they can initiate and regulate inflammatory processes by secreting the inflammatory mediator such as NO, as well as the pro-inflammatory cytokines such as TNF-α and IL-6 [2,11], so the effects of PFF on levels of NO and cytokines produced in RAW264.7 cell were investigated. Firstly, the cell viability under the PFF intervention was determined. As illustrated in Figure 1, PFF exhibited no cytotoxic effect on RAW264.7 cells with a concentration lower than 200 μg/mL, so three concentrations of PFF (50, 100, and 200 μg/mL) were selected for the following experiment. Then, NO, as a momentous free radical involved in the inflammation process [23], was determined. It was observed that PFF significantly inhibited NO production in LPS-induced RAW264.7 cells and lowered its level from 42.60 μM down to 7.70–23.51 μM at a concentration range of 50–200 µg/mL (*p* < 0.01) (Figure 2A). Meanwhile, as shown in Figure 2B, C, PFF treatment (50–200 µg/mL) efficiently suppressed the production of TNF-α and IL-6 in a dose-dependent manner (*p* < 0.01), and the inhibitory rates were 21.86–79.51% for TNF-α and 26.76–82.06% for IL-6. Therefore, from the above experimental results, it could be found that PFF exhibited anti-inflammation potential by suppressing some inflammatory factors.

### 3.2. Effects of PFF on MAPK, PIK/Akt, and NF-κB Signaling Pathways in RAW264.7 Cells

It has been reported that MAPK and PI3K/Akt pathways are closely associated with inflammatory reactions [31]. In particular, the phosphorylation of some extracellular signal kinases such as ERK, p38, JNK, and Akt related to these two pathways are the important indexes of inflammation [2]. By Western blot results, it could be found that the phosphorylation levels of ERK, p38, JNK, and Akt were significantly up-regulated in LPS-induced RAW264.7 cells (*p* < 0.01), which was consistent with the inflammatory indications. However, after intervention with PFF (50–200 μg/mL), the phosphorylation levels of p38, JNK, and Akt were significantly reduced in a dose-dependent manner (*p* < 0.01), and reduction rates were 21.35–61.47% for p38, 15.88–29.10% for JNK, and 5.18–50.24% for Akt (Figure 3B–D). Meanwhile, the phosphorylation level of ERK was not evidently changed after treatment with PFF in a concentration ranging from 50 to 100 μg/mL, but it was decreased significantly by 10.96% at doses of 200 μg/mL PFF (*p* < 0.01) (Figure 3A). Moreover, considering that NF-κB, a vital transcription factor mainly composed of IκB and p65, plays a significant role in regulating multiple inflammatory diseases [32], the impacts of PFF on NF-κB/p65 translocation were also studied. The results (Figure 4) showed that p65 protein was overexpressed in the nuclear in LPS-induced RAW264.7 cells; however, its expression was significantly reversed by PFF treatment (100 and 200 μg/mL), indicating that PFF disturbed the translocation of p65 stimulated by LPS. Collectively, the above results suggested that the anti-inflammatory activity of PFF was correlated with the activation of MAPK, PIK/Akt, and NF-κB signaling pathways.

### 3.3. Intestinal Regulatory Activity of PFF

#### 3.3.1. Effects of PFF on the Gut Microbiota

The increasing evidence showed that gut microbiota and their metabolites are closely correlated with inflammatory reactions in organisms [5,21]. In this work, an in vitro simulation fermentation by human fecal bacteria was used to assess the probiotic effects of PFF according to the BFBL gut model and explore the correlation between gut microbiota and intestinal inflammation. Thus, we determined the targeted bacterial count in three colon reactors (R_1_, R_2_, and R_3_) used to simulate the human colon (Table 1). Compared with the stabilization period (CK), 7-day PFF consumption could inhibit several potentially pathogenic bacteria, including *Alistipes*, *Bilophia*, *Enterobacteriaceae*, and so on, which are generally correlated with intestinal mucosal injury and inflammatory diseases [33,34,35]. However, the abundance of *Enterococcus*, *Lactobacillus*, and *Roseburia* was obviously promoted (*p* < 0.01). In addition, *Bifidobacterium* also tended to increase slightly. Fortunately, all these bacteria can strengthen the intestinal mucosal barrier and have a positive role in alleviating inflammation by producing SCFAs [35,36,37,38].

#### 3.3.2. Effects of PFF on the Metabolites

Considering some SCFA-producing bacteria were significantly promoted by PFF, we further assessed the SCFA production during the PFF intake period (Figure 5). The result showed that, during the stabilization period (CK), acetic acid ranked first in SCFA content, followed by propionic acid and butyric acid. Afterward, 7-day PFF consumption markedly increased the SCFA content in R_1_, R_2_, and R_3_ compared with the stabilization period (CK) (*p* < 0.05). Among these, the content of acetic acid, as the main component of SCFAs, increased by 1.19-, 1.03-, and 1.17-fold in R_1_, R_2_, and R_3_, respectively. These results indicated that PFF consumption was a good source or promoter of SCFA in gut microbiota. In addition, high levels of SCFAs are desirable because their corresponding pH reduction can inhibit the growth of harmful bacteria, thereby preventing colonic inflammation [39,40]. On the whole, our data suggested a potential role for PFF in attenuating intestinal inflammation, such as increasing the richness of specific beneficial bacteria and content of SCFAs, as well as inhibiting the growth of harmful bacteria.

### 3.4. Initial Analysis of the Main Flavonoids in PFF

To confirm the component basis of PFF on the effect of inflammation or gut microbiota, the bioactive components of PFF were further analyzed using spectroscopic methods. The result showed that PFF contained five main compounds isolated and identified as flavonoids by HPLC combined with UV-Vis and FT-IR spectra (Appendix A). Moreover, LC-MS data also showed that the excimer ion peaks [M − H]^−^ of the five flavonoid compounds were 609.14691, 593.15125, 563.14044, 609.14630, and 593.15112, respectively (Appendix A). Based on the above data and relevant references [27,32,41,42,43,44,45], the five compounds were preliminarily inferred as orientin-2″-*O*-galactoside, apigenin-6,8-di-*C*-glucoside, vitexin-2″-*O*-xyloside, orientin-2″-*O*-glucoside, and vitexin-2″-*O*-glucoside, respectively. Flavonoids and their derivatives can prevent and treat various diseases, including inflammatory diseases [46]. Previous studies reported that orientin could effectively alleviate inflammatory reactions in LPS-induced RAW 264.7 cells [47,48], vitexin could down-regulate inflammatory mediators and reduce the migration of neutrophils to the inflammatory focus [49], and apigenin could lighten the LPS-stimulated BV2 microglia by inhibiting the release of pro-inflammatory cytokines [50]. Hence, these five flavonoid compounds may play a key role in the anti-inflammatory activity exerted by PFF, but their structures and biological activities still require further analysis and determination. Overall, this study could promote the development of novel natural anti-inflammatory agents based on bioactive flavonoids.

## 4. Conclusions

In this study, PFF extracted from *P. foetida* fruits could effectively inhibit the biosynthesis and production of the inflammatory mediator (NO) and pro-inflammatory cytokines (TNF-α and IL-6) from relieving the symptoms of inflammatory responses. Moreover, it could also obviously down-regulate the phosphorylation levels of pro-inflammatory proteins (ERK, JNK, p38, Akt, and p65) involved in regulating MAPK, PI3K/Akt, and NF-κB signaling pathways. More importantly, PFF exhibited the growth-promoting ability of several beneficial bacteria, including *Bifidobacterium*, *Enterococcus*, *Lactobacillus*, and *Roseburia*, and SCFA generation ability in gut microbiota. These metabolism effects could be considered to alleviate inflammatory reactions. In addition, PFF mainly consists of five flavonoid compounds, which might be active compounds with anti-inflammatory properties. Therefore, flavonoids derived from *P. foetida* fruit, as new preventive and therapeutic substances, provide the possibility to develop functional components to alleviate inflammatory diseases in the food and pharmaceutical industries.

## Figures and Tables

**Figure 1 foods-12-02889-f001:**
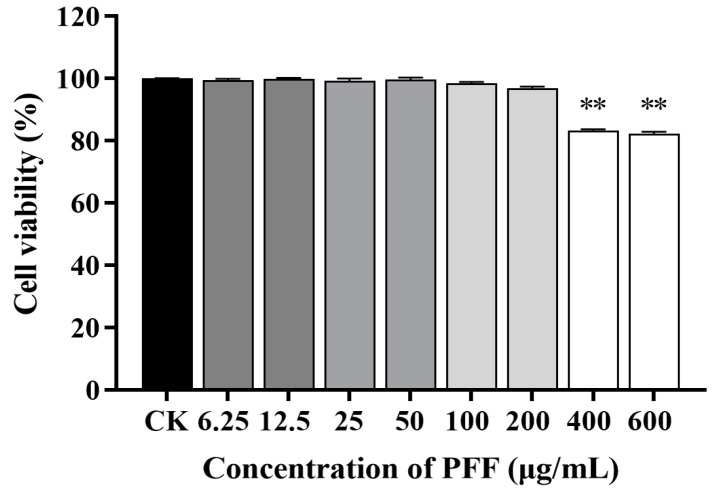
Effects of PFF on cell viability of RAW264.7 cells. Data were exhibited as mean ± SD. ** *p* < 0.01, vs. the control group.

**Figure 2 foods-12-02889-f002:**
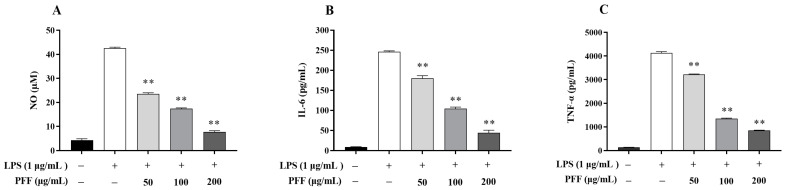
Effects of PFF on production of NO (**A**), IL-6 (**B**), and TNF-α (**C**) in LPS-induced RAW264.7 cells. Data were exhibited as mean ± SD. ** *p* < 0.01, vs. the LPS-treated group (without PFF).

**Figure 3 foods-12-02889-f003:**
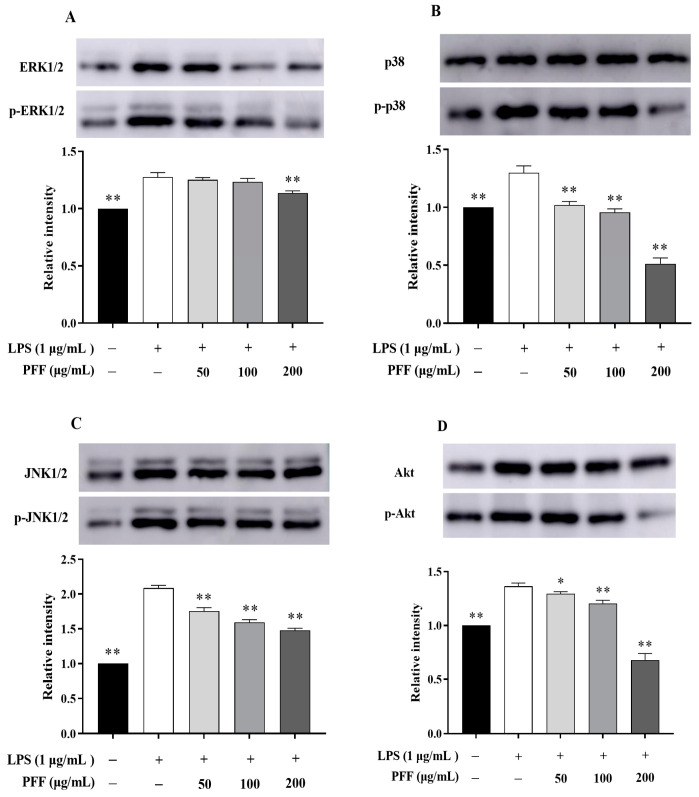
Effects of PFF on the expression of major immune proteins in MAPK and PI3K/Akt signaling pathways by western blotting. Representative immunoblots of p-ERK (**A**), p-JNK (**B**), p-p38 (**C**), and p-Akt (**D**) proteins are shown. Data were exhibited as mean ± SD. * *p* < 0.05, ** *p* < 0.01, vs. the LPS-treated group (without PFF).

**Figure 4 foods-12-02889-f004:**
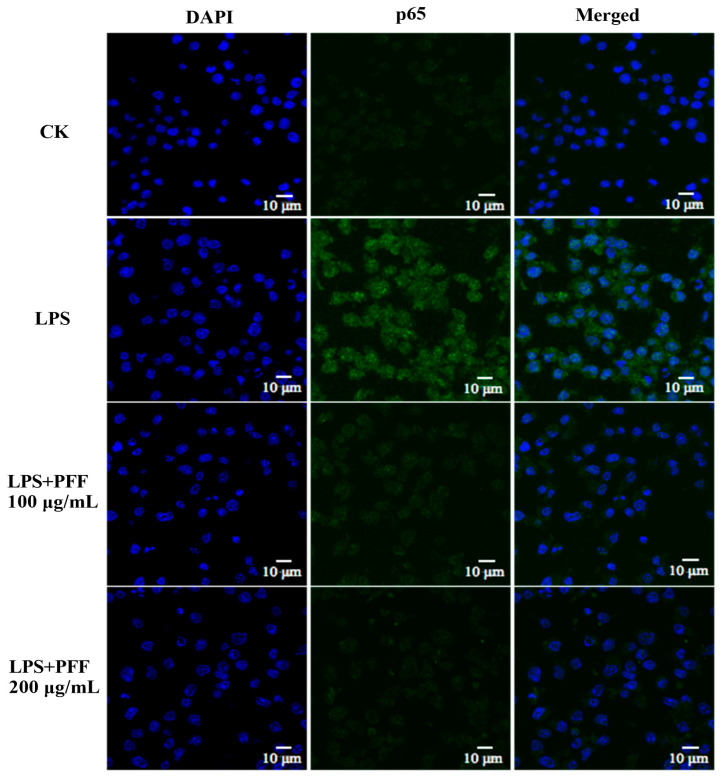
Effects of PFF on p65 nuclear translocation of LPS-stimulated RAW264.7 cells.

**Figure 5 foods-12-02889-f005:**
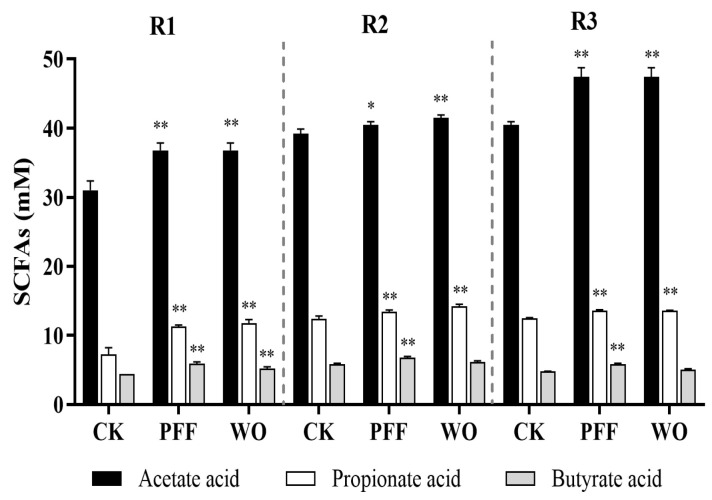
Effects of PFF on SCFAs content in the BFBL gut model. SCFAs content in the ascending colon (R_1_), transverse colon (R_2_), and descending colon (R_3_) of three experiment periods (CK, PFF, and WO). Data were exhibited as mean ± SD. * *p* < 0.05, ** *p* < 0.01, vs. the CK.

**Table 1 foods-12-02889-t001:** qPCR values (mean log copy number/mL ± SD) for targeted microbial group analysis in ascending colon (R_1_), transverse colon (R_2_), and descending colon (R_3_) of three experiment periods (CK, PFF, and WO). (* *p* < 0.05, ** *p* < 0.01, vs. the CK).

Bacterial Group	Compartment	CK	PFF	WO
*Alistipes*	R_1_	7.07 ± 0.01	5.31 ± 0.21 **	5.38 ± 0.06 **
	R_2_	8.69 ± 0.07	8.58 ± 0.14	8.36 ± 0.06 *
	R_3_	8.75 ± 0.03	8.45 ± 0.09 *	8.46 ± 0.14 *
*Bifidobacterium*	R_1_	8.84 ± 0.31	8.65 ± 0.12	8.36 ± 0.09
	R_2_	8.30 ± 0.38	8.65 ± 0.20	8.46 ± 0.08
	R_3_	8.30 ± 0.38	8.65 ± 0.20	8.12 ± 0.18
*Bilophia*	R_1_	8.37 ± 0.01	8.27 ± 0.10	8.22 ± 0.27
	R_2_	8.39 ± 0.01	8.16 ± 0.09	8.23 ± 0.24
	R_3_	8.38 ± 0.20	7.99 ± 0.20	8.30 ± 0.18
*Enterbacteriaceae*	R_1_	8.60 ± 0.00	8.26 ± 0.02 **	7.83 ± 0.05 **
	R_2_	8.49 ± 0.01	8.12 ± 0.05 **	7.90 ± 0.02 **
	R_3_	8.48 ± 0.05	7.91 ± 0.13 **	8.03 ± 0.04 **
*Enterococcus*	R_1_	6.48 ± 0.08	8.56 ± 0.21 **	8.87 ± 0.12 **
	R_2_	6.42 ± 0.13	8.84 ± 0.13 **	8.82 ± 0.01 **
	R_3_	7.09 ± 0.04	8.44 ± 0.21 **	8.40 ± 0.02 **
*Lactobacillus*	R_1_	3.68 ± 0.00	3.99 ± 0.02 **	5.54 ± 0.04 **
	R_2_	3.60 ± 0.03	4.44 ± 0.15 **	5.41 ± 0.00 **
	R_3_	3.98 ± 0.00	4.42 ± 0.07 **	5.28 ± 0.05 **
*Roseburia*	R_1_	5.64 ± 0.09	6.47 ± 0.19 **	4.93 ± 0.02 **
	R_2_	6.23 ± 0.07	6.58 ± 0.06 **	5.56 ± 0.03 **
	R_3_	6.01 ± 0.03	6.47 ± 0.10 **	5.29 ± 0.06 **

## Data Availability

All data are reported in the article.

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
