# Peer review of "Anti-Inflammatory and Gut Microbiota Modulation Potentials of Flavonoids Extracted from Passiflora foetida Fruits"

_foods, 2023, doi:10.3390/foods12152889_

Round 1
Reviewer 1 Report
Thank you for submitting your work. In general, it is a high-quality work in terms of biochemical analysis, however, the flavonoid fraction analysis needs improvement and clarification. My comments are provided on the manuscript in the attached file.

Author Response
Response to Reviewer 1 Comments
Dear Reviewer,
I want to express my appreciation to you for providing excellent and detailed suggestions and comments on our manuscript. These comments are all valuable and helpful for improving our manuscript. We have carefully revised the manuscript and responded to each of the points you raised. In this revised version, the main changes to our manuscript were highlighted within the document using red-colored text. We really appreciate your kindness and help. Thank you very much!
Point 1: Please add further information about the nutrition composition of the fruit, including the quantities of polyphenolic compounds. (Lines 70-71)
Response 1: Thank you for your comment. We have added further information on the nutritional composition of Passiflora foetida fruits in lines 70-72.
Point 2: The name needs to be in italic in the entire text. (Line 81)
Response 2: Thank you for your careful review. We have modified it throughout the entire manuscript.
Point 3: When was the harvest season? (Line 81)
Response 3: Thank you very much. We have added detailed information in line 83.
Point 4: Did you quantify the flavonoid content? It is important to understand the concentration of these compounds in the tested extract. (Lines 96-105)
Response 4: Thank you for raising this very important issue. We found that flavonoids were the main compounds in PFF, and the total flavonoid content in PFF was 254 µg/mg as determined by the aluminum nitrate method. Based on HPLC and LC-MS analysis, we also preliminarily inferred that PFF mainly contained five compounds, which were orientin-2″-O-galactoside, apigenin-6,8-di-C-glucoside, vitexin-2″-O-xyloside, orien-tin-2″-O-glucoside, and vitexin-2″-O-glucoside, respectively. In the follow-up work, we will further isolate and identify these five compounds to quantify their contents accurately.
Point 5: What are the references for the selected analysis methods? (Line 201)
Response 5: Thank you for your careful review of our manuscript. We have supplemented the references for the selected analytical methods in Section 2.10. The relevant references are as follows:
- Wang, Y.; Zhang, Y.; Hou, M.; Han, W. Anti-fatigue activity of parsley (Petroselinum crispum) flavonoids via regulation of oxidative stress and gut microbiota in mice. J Funct Foods. 2022, 89, 104963.
- Xu, X.; Xie, H.; Hao, J.; Jiang, Y.; Wei, X. Flavonoid glycosides from the seeds of Litchi chinensis. Agric. Food Chem. 2011, 59, 1205-1209.
Point 6: This section sounds confusing to the reader, please further divide this section into sub-sections outline each type of analysis separately + reference. (Lines 214-216)
Response 6: Thank you for your comments. We have revised it according to your suggestion. (Lines 210-220)
Point 7: What was the purpose of this analysis? How was it performed? What outcomes were you targeting? (Lines 214-216)
Response 7: Thank you for raising these important issues. Our preliminary study found that flavonoid-rich fraction (PFF) extracted from P. foetida fruits had excellent anti-inflammatory activities. Based on this, further investigation is necessary to explore the composition basis of PFF to reveal the active flavonoids mainly contained in PFF. The above work will lay a foundation for further exploring flavonoids' composition, structure, and anti-inflammatory mechanism.
Point 8: This section requires further improvement and clarification. (Line 309)
Response 8: Thank you for your comments. The content we added in the manuscript is as follows: The result showed that PFF contained five main compounds isolated and identified as flavonoids by HPLC combined with UV-Vis and FT-IR spectra (Figures S2-S4). Moreover, LC-MS data also showed that the excimer ion peaks [M-H]- of the five flavonoid compounds were 609.14691, 593.15125, 563.14044, 609.14630, and 593.15112, respectively (Figure S5). Based on the above data and relevant references[27,32,41-45], the five compounds were preliminarily inferred as orientin-2″-O-galactoside, apigenin-6,8-di-C-glucoside, vitexin-2″-O-xyloside, orientin-2″-O-glucoside, and vitexin-2″-O-glucoside, respectively.
Point 9: I am not able to view the supplementary files. The format of the folder is not recognized. Please provide it as a PDF. (Line 314)
Response 9: Thank you very much. We have added figures S2-S5 in the supplementary materials to this Word document.
Point 10: On what basis did you decide that these are main compounds? Did you quantify them? (Lines 313-315)
Response 10: Thank you for raising this very important issue. We performed a semi-quantitative study of PFF by HPLC and LC-MS (Figures S2-S5). By HPLC analysis, we found that PFF mainly contained five compounds with high relative content (Figure S2). Therefore, we further used LC-MS and combined with relevant references analysis to preliminarily infer that the above five compounds were orientin-2″-O-galactoside, apigenin-6,8-di-C-glucoside, vitexin-2″-O-xyloside, orien-tin-2″-O-glucoside, and vitexin-2″-O-glucoside, respectively (Figure S5). In addition, we are also in the process of separating and purifying the above five compounds, and the follow-up work will confirm the above flavonoids by NMR and other means.

Reviewer 2 Report
In this article, the authors studied the anti-inflammatory and intestinal microbiota modulation potentials of the flavonoid-rich fraction (PFF) extracted from the fruits of Passiflora foetida, demonstrating its efficacy in inhibiting the biosynthesis and production of the inflammatory mediator (NO ) and pro-inflammatory cytokines (TNF-α and IL-6)
However, there are some comments, curiosity and suggestions that I want to share.
How PFF promotes the growth of different lactic acid bacteria?
The authors should better explain the experiments performed to determine that Bifidobacterium, Enterococcus, Lactobacillus, and Roseburia have had improvements using PFF (growth curves or viability or final OD600)
Author Response
Response to Reviewer 2 Comments
Dear Reviewer,
I want to express my appreciation to you for providing excellent and detailed suggestions and comments on our manuscript. These comments are all valuable and helpful for improving our manuscript. We have carefully revised the manuscript and responded to each of the points you raised. In this revised version, the main changes to our manuscript were highlighted within the document using red-colored text. We really appreciate your kindness and help. Thank you very much!
Point 1: How PFF promotes the growth of different lactic acid bacteria? The authors should better explain the experiments performed to determine that Bifidobacterium, Enterococcus, Lactobacillus, and Roseburia have had improvements using PFF (growth curves or viability or final OD600).
Response 1: Thank you for raising this issue. In this study, we found that the flavonoid-rich fraction (PFF) extracted from Passiflora foetida fruits possessed immune-enhancing and anti-inflammatory activities. However, many studies have shown that most beneficial substances (such as flavonoids) cannot be degraded and absorbed in the stomach and small intestine and need to be hydrolyzed by gut microbiota in the large intestine to produce metabolites (such as short-chain fatty acids), to play an active role. Thus, the effects of PFF on gut microbiota and its metabolites need to be explored. In recent years, the establishment of in vitro continuous dynamic intestinal simulation digestive system based on human physiology, such as SHIME, TIM-1/TIM-2, and SIMGI, are more convenient, efficient, and low-cost than in vivo studies, which provides convenience for studying the digestive characteristics and bioavailability of beneficial substances such as polysaccharides and polyphenols. The BFBL system used in our study was established based on SHIME, TIM-1/TIM-2, and SIMGI. These simulation systems have been proven to be suitable for evaluating the impact of food ingredients on regulating human gut microbiota (References 1-3). In the complex intestinal microenvironment system, it is necessary to consider the interaction between various bacteria. Therefore, we used real-time fluorescent quantitative PCR to quantify the relative abundance of gut microbiota and explore the effect of PFF on them. We performed the experiment according to the previous methods (References 4-6), with a few modifications. The above references are as follows:
- Van den Abbeele, P.; Venema, K.; Van de Wiele, T.; Verstraete, W.; Possemiers, S. Different human gut models reveal the distinct fermentation patterns of Arabinoxylan versus inulin. Agric. Food Chem. 2013, 61, 9819-9827.
- Von Martels, J.; Sadaghian, S. M.; Bourgonje, A. R.; Blokzijl, T.; Dijkstra, G.; Faber, K. N.; Harmsen, H. The role of gut microbiota in health and disease: In vitro modeling of host-microbe interactions at the aerobe-anaerobe interphase of the human gut. Anaerobe, 2017, 44, 3-12.
- Cardenas-Castro, A. P.; Bianchi, F.; Tallarico-Adorno, M. A.; Montalvo-Gonzalez, E.; Sayago-Ayerdi, S. G.; Sivieri, K. In vitro colonic fermentation of Mexican "taco" from corn-tortilla and black beans in a Simulator of Human Microbial Ecosystem (SHIME®) system. Food Res. Int. 2019, 118, 81-88.
- Barroso, E.; Montilla, A.; Corzo, N.; Peláez, C.; Martínez-Cuesta, M. C.; Requena, T. Effect of lactulose-derived oligosaccharides on intestinal microbiota during the shift between media with different energy contents. Food Res. Int. 2016, 89, 302-308.
- Requena, T.; Song, Y.; Peláez, C.; Martínez-Cuesta, M. C. Modulation and metabolism of obesity-associated microbiota in a dynamic simulator of the human gut microbiota. LWT. 2021, 141, 110921.
- Zhu, M.; Song, Y.; Martínez-Cuesta, M. C.; Peláez, C.; Li, E.; Requena, T.; Wang, H.; Sun, Y. Immunological activity and gut microbiota modulation of pectin from kiwano (Cucumis metuliferus) peels. Foods. 2022, 11, 1632.

Round 2
Reviewer 1 Report
Thanks for revising the manuscript.
Point 1 has not been addressed properly. Giving the number of nutrients (e.g., number of amino acids and minerals) does not benefit the readers. The main nutrients need to be specified with quantities.
Author Response
Response to Reviewer 1 Comments
Dear Reviewer,
I want to express my appreciation to you for providing excellent and detailed suggestions and comments on our manuscript. These comments are all valuable and helpful for improving our manuscript. We have carefully revised the manuscript and responded to each of the points you raised. In this revised version, the main changes to our manuscript were highlighted within the document using red-colored text. We really appreciate your kindness and help. Thank you and best regards.
Point 1: Point 1 has not been addressed properly. Giving the number of nutrients (e.g., number of amino acids and minerals) does not benefit the readers. The main nutrients need to be specified with quantities.
Response 1: Thank you for your comment. We have revised it according to your suggestion. The content we added to the manuscript is as follows:
Our previous studies showed that P. foetida fruits are mainly rich in amino acids (1097 mg/100 g), minerals (595.75 mg/100 g), sugars (3.34 g/100 g), organic acids (0.24 mg/100 g), and polyphenols (284.4 mg/100 g), especially flavonoids (239 mg/100 g), which are the major component of polyphenols[18]. (Lines 70-73)
